# Anti-Markovnikov hydro(amino)alkylation of vinylarenes via photoredox catalysis

Zhao Wu [1,3], Samuel N. Gockel[1,2,3] & Kami L. Hull [2✉]

Photoredox catalysis is a powerful means to generate odd-electron species under mild reaction conditions from a wide array of radical precursors. Herein, we present the application of this powerful catalytic manifold to address the hydroalkylation and hydroaminoalkylation of electronically diverse vinylarenes. This reaction allows for generalized alkene hydroalkylation leveraging common alkyl radical precursors, such as organotrifluoroborate salts and carboxylic acids. Furthermore, utilizing easily accessible α-silyl amine reagents or tertiary amines directly, secondary and tertiary amine moieties can be installed onto monoaryl and diaryl alkenes to access valuable products, including γ,γ-diarylamines pharmacophores. Thus, under a unified system, both hydroalkylation and hydroaminoalkylation of alkenes are achieved. The substrate scope is evaluated through 57 examples, the synthetic utility of the method is demonstrated, and preliminary mechanistic insights are presented.

[1] Department of Chemistry, University of Illinois at Urbana-Champaign, 600 South Mathews Avenue, Urbana, IL 61801, USA. [2] Department of Chemistry, University of Texas at Austin, 100 East 24th Street, Austin, TX 78712, USA. [3] These authors contributed equally: Zhao Wu, Samuel N. Gockel. ✉email: kamihull@utexas.edu

Olefin functionalization is a routinely applied strategy in organic synthesis[1]. Feedstock alkenes such as vinylarenes are attractive templates for a variety of reaction manifolds, including polar and radical reactions, cycloadditions, and transition metal-catalyzed functionalizations[2]. In particular, the conversion of the π-bond into polar groups, such as alcohols[3], ethers[4], thioethers[5,6], amines[7,8], halogens[9], or carbonyls[10,11] has seen substantial development in recent decades. The identification of such desirable functional group targets in medicinal chemistry has been an important motivating force in the development of these methods[12].

In evaluating the occurrence of functionalities found broadly in organic chemistry, alkylated arene fragments emerge as prominent features in many important classes of molecules, particularly in pharmaceutical agents[13]. Hydro(amino)alkylation reactions offer a complementarity for the synthesis of these important moieties to traditional alkyl–alkyl cross-couplings catalyzed by transition metals for the synthesis of these important moieties (Fig. 1)[14,15]. First, the requirement for water-sensitive organometallic coupling partners, such as Grignard reagents, organolithiums, zincates, or cuprates can be avoided, expanding overall practicality[16,17]. In addition, kinetic challenges that commonly plague transition metal catalysis, such as sluggish activation of alkyl halides or competitive β-hydride elimination, can be entirely circumvented[18]. Given this complementarity, the development and refinement of general and catalytic platforms for alkene hydroalkylations is of significant interest.

Common to any olefin functionalization process, the regioselectivity of the addition is a critical parameter. The majority of transition metal-catalyzed hydroalkylation reactions, such as those promoted by Pd[19], Ni[20–22], and Cu[23], initiate with a hydrometallation step, which engenders exclusive Markovnikov selectivity with vinylarenes given the stability of the π-benzyl intermediate (Fig. 1c, top left). Correspondingly, anti-Markovnikov hydroalkylation of styrenes remains underdeveloped. Carbometallation-initiated strategies have emerged to address this regiochemical challenge[24,25]. However, the majority of this methodology is applicable only to electronically activated Michael acceptors. When applied to vinylarenes, reactive organometallic (Mg[26], Li[27,28], K[29]) reagents are required, and chemical yields vary significantly with the electron properties of the π-system (Fig. 1c, top right). Anti-Markovnikov hydroaminoalkylations of vinylarenes have been met with greater success, albeit requiring privileged ligand scaffolds and high pressures of syngas to promote reactivity and regioselectivity (Fig. 1c, bottom left)[30–35]. Alternatively, these aliphatic amines can be accessed through intermolecular anti-Markovnikov hydroamination, a complementary disconnection, which has been a great success recently using photoredox catalysis[36,37].

Anti-Markovnikov-selective hydro(amino)alkylations of electron-deficient alkenes initiating with the addition of a carbon-centered radical to the π-system have been well-developed. However, given the nucleophilic character of (amino)alkyl radicals, such reactions have largely been restricted to highly polarized alkenes, such as Michael acceptors[38–44]. To date, few select examples have emerged that demonstrate reactivity with vinylarenes. The scope is entirely restricted to highly nucleophilic radicals[45,46] or electrophilic styrenes[47–50], strongly reducing conditions[51], or by stoichiometric activation of the arene by metal carbonyl complexes (Fig. 1c, bottom right)[52]. At present, a system that exhibits a generality of scope under practical conditions remains unknown.

Herein, we report a unified catalytic system for anti-Markovnikov hydroalkylation and hydroaminoalkylation of various vinylarenes through photoredox catalysis. We demonstrate that the modular access to high-value alkylated arene fragments is readily accessed via the intermolecular hydroalkylation or hydroaminoalkylation of feedstock vinylarenes (Fig. 1a). This strategy enables the direct synthesis of valuable amine pharmacophores well-represented in top-selling pharmaceuticals like Sensipar, Milverine, or Avil and their derivatives[53].

## Results

**Reaction discovery and optimization.** To address this important synthetic gap in the otherwise well-developed suite of hydroalkylation reactions, we sought to harness the regioselectivity of a single-electron mechanism. Visible light photocatalysis has been established as a means for the facile production of the requisite open-shell alkyl species from a diverse array of precursors[54]. Oxidizable substrates, such as organoborons, organosilicons, or carboxylic acids are a particularly attractive suite of reagents; these inexpensive and shelf-stable substrates undergo facile activation via reductive quenching of common photoredox catalysts. In accordance with our proposed mechanistic outline, the addition of the generated radical to the vinylarene generates a benzylic radical (Fig. 2). Processing of this intermediate by stepwise or concerted electron and proton transfer would deliver the hydroalkylated product. However, such intermediates tend to have large negative reduction potentials ($E = -1.60$ V vs. saturated calomel electrode (SCE))[55]; identification of a hydrogen atom transfer reagent, therefore, represents a key reaction parameter.

Considering these design principles, we initiated our study of the hydroalkylation reaction of 1,1-diphenylethylene with benzyl potassium trifluoroborate to form **1**. Surveying common H-sources, alcohols were found to perform poorly in this hydroalkylation reaction (Table 1, Entries 1–2). Thiophenol, a common H-atom transfer reagent, produces the thiol-ene adduct exclusively (Table 1, Entry 4). Ultimately, phenol derivatives were found to be uniquely effective (Table 1, Entries 5–13). This could be a consequence of the lower $pK_a$ or the lower O–H bond dissociation energy, depending on the mode of H-transfer (*vide infra*). Alternatively, it could be related to the ability of the phenoxide anion to capture liberated $BF_3$, which is known to inhibit similar transformations[56,57]. An excess of the phenol reagent is found to be beneficial (Table 1, Entry 6).

Additional tuning of the phenol reagent and reaction time led to the optimized conditions (Table 1, Entry 9): 1.0 equivalent of the alkene, 1.5 equivalents of the alkyl-$BF_3K$ salt undergo the coupling in 48 h with 3.0 equivalents of 2-methoxyphenol and 1 mol% of $[Ir(dF(CF_3)ppy)_2(dtbbpy)][PF_6]$ in dimethylformamide (DMF) (0.10 M) with irradiation by 24 W blue light-emitting diodes (LEDs). Notably, other phenol derivatives were found to provide good to excellent yields of 1 (Table 1, Entries 10–13); however, 2-methoxyphenol (guaiacol) is a naturally occurring and inexpensive feedstock.

**Substrate scope.** With optimized conditions in hand, the scope of hydroalkylation reaction was investigated (Fig. 3). Both 1° and 2° alkyl trifluoroborate salts (**1–8**) are reactive to afford the desired products in good yields. The efficiency of n-butyl-$BF_3K$ (**4**) is noteworthy due to the significantly higher oxidation potential of this substrate (+1.81 V vs. SCE for phenethyl-$BF_3K$ vs. +1.05 V vs. SCE for Bn-$BF_3K$)[48]. With respect to the trifluoroborate carbon components, the reaction efficiency was found to be sensitive to the structure of the phenol derivative. For instance, **3** can be obtained in good yield with 2-methoxyphenol (79%); however, phenol itself was found to be more effective, providing 98% yield. Nitrogen-containing alkyl fragments can be incorporated into the products utilizing 4-methoxyphenol, such as the 1-boc-piperidinyl moiety (**6**). Formal hydroaminoalkylation can also be achieved α-amino-$BF_3K$ salts (**16–17**). In addition to benzylic moieties (**1–2**),

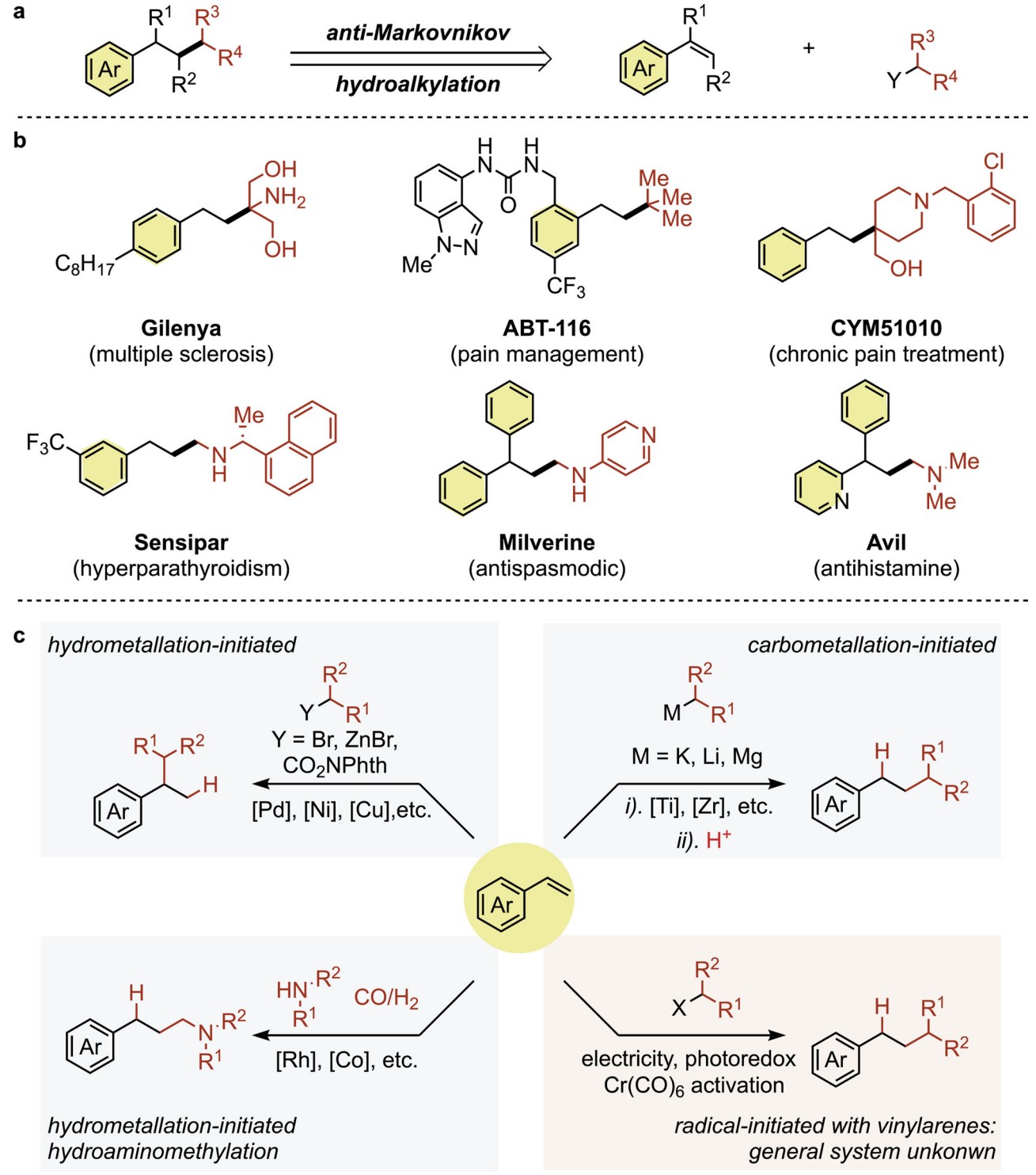

**Fig. 1 Significance and syntheses of (amino)alkylated arenes. a** Accessing valuable (amino)alkylated arenes via anti-Markovnikov hydro(amino)alkylation of simple styrene derivatives. **b** Biologically active molecules containing (γ-amino) alkylated arenes. **c** Literature reports for hydroalkylation of styrenes: hydrometallation-initiated Markovnikov hydroalkylation; carbometallation-initiated anti-Markovnikov hydroalkylation; hydrometallation-initiated anti-Markovnikov hydroaminomethylation using syngas; and radical-initiated anti-Markovnikov hydroalkylation.

allylic fragments can also be installed (**7**). The synthesis of **7** was particularly sensitive to the phenol equivalencies, with lower loadings proving more effective (see Supplementary information for details). This could be a consequence of a slower radical addition by the stabilized allylic species. Importantly, the alkene remains intact, leaving this valuable functionality available for further elaboration. Steric hindrance in the incorporated alkyl

fragment is tolerated, as demonstrated by **8**, which is furnished in 74% yield with 5:1 diastereomeric ratio (d.r.).

Alkyl potassium trifluoroborate salts are versatile substrates for this reaction. However, we sought to expand substrate generality to include other common radical precursors. Alkyl carboxylic acids are an abundant feedstock and the ability to engage these directly would greatly enhance the practicality and utility of the

transformation. With these substrates, external H-sources are not required, and the conditions could be easily adjusted. An evaluation of carboxylic acid substrates reveals that secondary and tertiary carboxylic acids are more reactive under optimized conditions compared to primary ones (e.g., **3** vs. **4**), likely due to the lower oxidation potential of the substrate and the more stabilized alkyl radical intermediate that is formed. Tertiary acids can be used to construct fully substituted carbon centers, which can be challenging to assemble by other means (**9–12**, **14–15**). Importantly, electron-rich monoaryl alkenes, which fail in related reactions, readily participate in this hydroalkylation protocol (**10–11**). The incorporation of the phenethyl group into a substrate is readily achieved with styrene and unactivated alkyl fragments (**12**). Finally, gemfibrozil and tranexamic acid both couple readily with mono- and diarylalkenes to afford **13**-**15**, demonstrating the tolerance of aryl ethers and secondary carbamates in the hydroalkylation reaction.

To expand the purview of this hydroalkylation platform to generally access aminoalkylated products, we investigated the

reactivity of tertiary amines. Such substrates are known to undergo a H-atom shift following SET oxidation to form the α-amino radical at the least hindered α position[58]. Following minor adjustments to the reaction conditions, tertiary aliphatic amines including diisopropylmethylamine (**18**), diisopropylethylamine (Hünig's base, (±)-**19**), trimethylamine (**20**), dicyclohexylmethylamine (±)-**24**, phenyldimethylamine (**23**), and cyclic (±)-**25** anilines all afford the hydroaminoalkylation products with very good yields. Biologically active diisopromine (**18**), pheniramine (±)-**21**, and tolpropamine (±)-**22** were all synthesized using this method.

The ability to use tertiary amines directly significantly expands the hydroaminoalkylation scope. However, only tertiary amines are reactive under the optimized conditions, and an excess of the amine substrate (2 equiv) paired with long reaction times (≥24 h) are required to achieve high conversion. This is likely due to the relatively high oxidation potential of 3° alkyl amines (>+1.0 V vs. SCE), which is even higher for 2° alkyl amines (>+1.5 V vs. SCE)[59]. The highly oxidizing excited state of the catalyst, [Ir(dF(CF$_3$)ppy)$_2$(dtbbpy)$^+$]*, also oxidizes the products in some cases. For example, when triethylamine was utilized under standard conditions, both the desired tertiary amine product and dealkylated secondary amine side products were obtained, complicating the purification process.

To overcome this constraint, we envisioned that the introduction of a silyl group alpha to the amine, which leads to a significant decrease in the oxidation potential (+0.71 V for α-TMS methylpiperidine, compared to +1.31 V for N-benzylpiperidine vs. SCE)[60], would facilitate SET to the photoexcited catalyst thereby increasing the reaction rate. This redox auxiliary approach has been extensively studied in modern electroorganic synthesis[61]. Indeed, when applying MeOH as proton source, this SET process is further promoted (see Stern–Volmer quenching experiments in the Supplementary information). Moreover, a less oxidizing photocatalyst can be applied, eliminating the unproductive dealkylation pathway. Thus, a significantly broadened scope for hydroaminoalkylation can be achieved with α-trimethylsilyl (TMS) amines, including the introduction of a secondary amine moiety into the products.

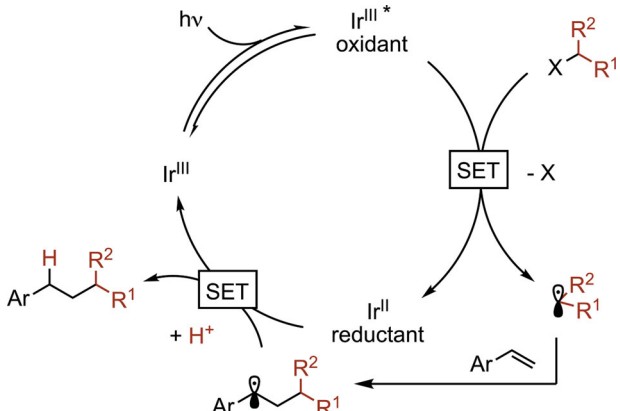

**Fig. 2 Mechanistic working hypothesis.** A proposed catalytic cycle for a general hydroalkylation of vinylarenes.

---

**Table 1 Select optimization results for the hydroalkylation reaction[a].**

Conditions scheme: 1,1-diphenylethylene (1.0 equiv) + KF$_3$B-CH$_2$-Ph (1.5 equiv), [Ir(dF(CF$_3$)ppy)$_2$(dtbbpy)][PF$_6$] (1.0 mol %), H-source (1.0-3.0 equiv), DMF, blue LED, rt → product **1**

| Entry | H-source | Equiv. H-source | t (h) | In situ yield 1 (%) |
|---|---|---|---|---|
| 1 | IPA | 3 | 24 | 16 |
| 2 | MeOH | 3 | 24 | 11 |
| 3 | AcOH | 3 | 24 | 15 |
| 4 | PhSH | 1 | 24 | 0 |
| 5 | PhOH | 1 | 24 | 16 |
| 6 | PhOH | 3 | 24 | 23 |
| 7 | PhOH | 3 | 36 | 42 |
| 8 | 2-MeO-PhOH | 3 | 36 | 84 |
| 9 | 2-MeO-PhOH | 3 | 48 | >99 |
| 10 | 4-CF$_3$-PhOH | 3 | 48 | 51 |
| 11 | 4-Cl-PhOH | 3 | 48 | 70 |
| 12 | 4-Me-PhOH | 3 | 48 | >99 |
| 13 | 4-MeO-PhOH | 3 | 48 | >99 |

[a]Conditions: 1,1-diphenylethylene (0.1 mmol, 1.0 equiv), BnBF$_3$K (1.5 equiv), [Ir(dF(CF$_3$)ppy)$_2$(dtbbpy)][PF$_6$] (1 mol%), H-source, DMF, r.t., 24 W blue LED light; GC yield reported with reference to 1-methylnaphthalene as an internal standard.

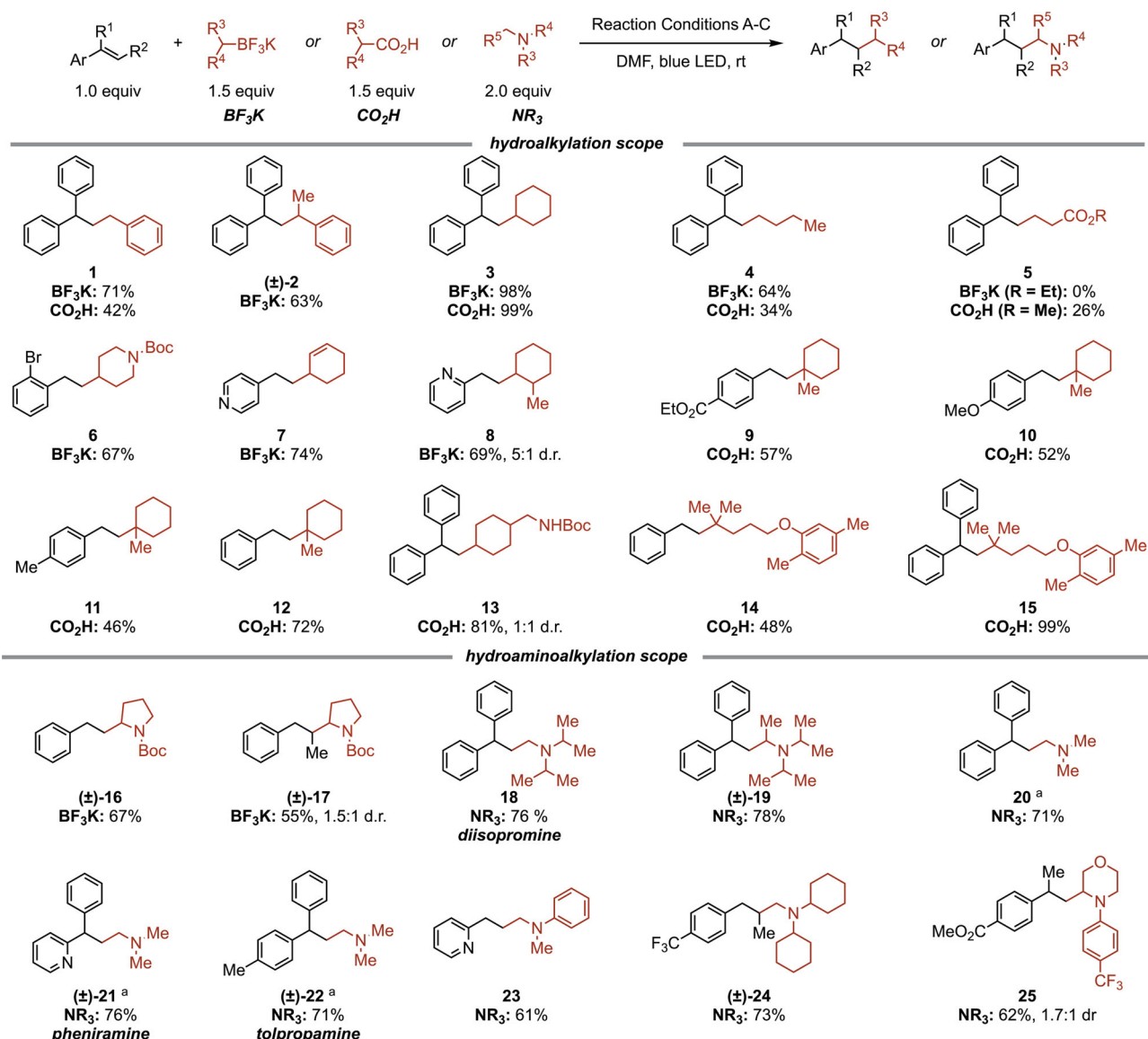

**Fig. 3 Scope of hydro(amino)alkylation.** Conditions A: trifluoroborate salt (1.5 equiv), PC (1.0 mol%), 2-methoxylphenol (3.0 equiv), DMF (0.1 M), r.t., 48 h; Conditions B: carboxylic acid (1.5 equiv), PC (1.0 mol%), K$_3$PO4 (1.5 equiv), DMF (0.2 M), r.t., 24 h; Conditions C: tertiary amine (2.0 equiv), PC (0.5 mol%), DMF (0.2 M), r.t., 24 h. **a** Me$_3$N·HCl (3.0 equiv) and DBU (3.0 equiv) were used. See Supplementary Methods for experimental details. d.r. diastereomeric ratio, PC [Ir(dF(CF$_3$)ppy)$_2$(dtbbpy)][PF$_6$].

As shown in Fig. 4, α-TMS amines derived from 2° cyclic and acyclic dialkyl amines, such as piperidine (**26**) or *N,N*-dibenzylamine (**31**) all afford the desired tertiary amine products in excellent yields. A variety of potentially sensitive functionalities were well-tolerated, including Boc-protected amine (**27**), basic tertiary amine (**28**), pyrimidine (**29**), and α-amino ester (**30**). Secondary α-TMS amines also proved to be effective under these conditions to give the 2° amine products in yields of 85% and 80% with the N–H bonds remaining intact (**32**, **33**).

For the alkene scope, various electronically differentiated 1,1-diaryl alkenes were evaluated. Electron-rich derivatives, such as 4,4′-dimethoxylphenylethylene, require a longer reaction time (24 h) to reach completion (**35**) with diminished yield (69%); however, hydroaminomethylation of electron-poor 4,4′-difluorophenylethylene (**34**) only requires 2 h for full conversion with an 84% isolated yield. Various monoaryl alkene derivatives were also examined. Vinylarenes bearing electron-withdrawing groups such as fluoro and bromo (**38**), trifluoromethyl (**39**), and ketone (**40**),

as well as electron-donating methyl (**42**) and methoxy (**43**, **44**) groups, all afford the desired amine products with moderate to excellent yields (31–84%). Heterocycles, such as thiazole (**45**) and pyridine (*ortho*, *meta*, and *para*, **46–48**) are well-tolerated under standard conditions for the synthesis of either secondary or tertiary amine products. Moreover, aryl, alkyl-disubstituted alkenes are amenable to the functionalization to access β- or γ-branched amine products (**49–51**) in good yields (54–72%). Finally, aryl alkenes derived from natural products, such as tryptamine (**52**), mycophenolic acid (**53**), and α-D-galactopyranose (**54**) are reactive under standard conditions to form the desired products with many functional groups remaining unaffected, including 2° amides, indoles, esters, lactones, internal alkenes, and acetals.

**Synthetic application**. Having developed a general methodology that enables hydroalkylation and hydroaminoalkylation of diverse

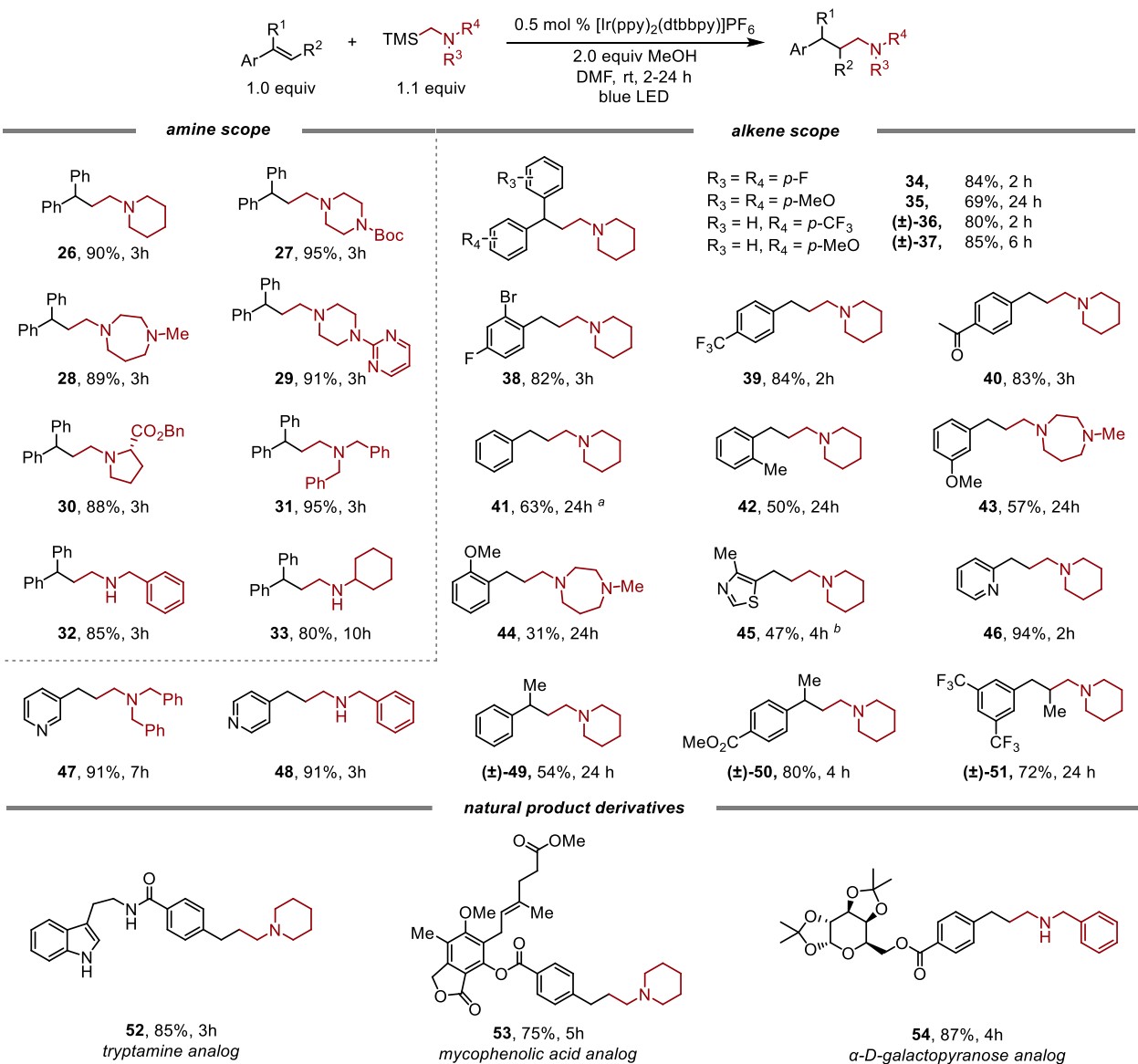

**Fig. 4 Scope of hydroaminomethylation with α-TMS amine.** Standard conditions (Conditions D) are alkene (0.2 mmol, 1.0 equiv), amine (1.1 equiv), [Ir(ppy)₂(dtbbpy)]PF₆ (0.5 mol%), DMF (1.0 mL, 0.1 M), r.t., 24 W blue LED light; isolated yields, average of two runs. ᵃThe 2.0 equiv amine used. ᵇ1.0 equiv amine and 2.0 equiv alkene used.

vinylarene substrates, we sought to demonstrate its application in the syntheses of biologically active γ-arylated amines as illustrated in Fig. 5. Starting from readily available α-TMS amines, synthesized via nucleophilic substitution of the corresponding chloromethyl silane, several γ-diphenyl amines were obtained in excellent yield (>80%) on gram-scale with only 0.1 mol% catalyst loading, thus demonstrating the synthetic potential of this method. It is worth noting that phenpyramine, a challenging substrate for Rh-catalyzed hydroaminomethylation (35% yield)[35], was prepared in 87% yield in 2.0 mmol scale employing our method. Sensipar, a top-selling pharmaceutical agent, is obtained from readily accessible *m*-CF₃ styrene and corresponding α-TMS amine utilizing this method in 3 mmol scale, without any erosion of enantiopurity.

**Mechanism investigations.** To provide support for our working mechanistic hypothesis outlined above (Fig. 2), a series of control reactions were carried out under optimized conditions for the hydroaminoalkylation (Fig. 6). In the presence of 2,2,6,6-

tetramethylpiperidinyloxy (TEMPO), the reaction was found to be completely inhibited, and the expected TEMPO-alkyl adduct was detected. To confirm the source of the H-atom in the product, deuterium-labeling studies were carried out. When the reaction is conducted in DMF-*d₇*, no deuterium incorporation at the benzylic site is observed, confirming that H-atom transfer from the solvent is not operative. To corroborate this result, the application of MeOD under the standard conditions leads to a product with 93% D incorporation at the benzylic position.

In the case of diphenylethylene studied above, the reduction potential of the radical intermediate ($E = -1.34$ V vs. SCE) is within range of the Ir photocatalyst ($E = -1.37$ V vs. SCE), which supports a sequential ET/PT mechanism involving proton transfer following prior reduction to the carbanion. However, with monoaryl alkene substrates, the benzylic radicals feature more negative reduction potentials that fall outside the redox window of the photocatalyst ($E \approx -1.45$ V vs. SCE for the styrene-derived intermediate). Although slightly endergonic, this electron transfer could be kinetically feasible if cage escape and

**Fig. 5 Synthetic application hydroaminoalkylaltion of vinylarenes.** Synthese of four bioactive molecules using this photoredox method with low catalyst loadings. PC [Ir(ppy)$_2$(dtbbpy)]PF$_6$ (photocatalyst). See Supplementary Methods for details.

**Fig. 6 Mechanistic investigations. a** Control experiments with TEMPO. **b** Deuterium-labeling studies in DMF-$d_7$. **c** Deuterium-labeling studies with MeOD. See Fig. 4 legend for Conditions D.

protonation of the subsequently formed carbanion competes with back electron transfer[62,63]. Alternatively, a concerted ET/PT mechanism[64] could render this step thermodynamically feasible. Although either mechanism cannot be strictly refuted based on current evidence, it is likely that different mechanisms may be operative for different substrates based on electronic properties. Similar mechanistic studies were conducted for the hydroalkylation reaction with carboxylic acids and provide additional supporting evidence (see Supplementary information for more details).

## Discussion

The anti-Markovnikov hydro(amino)alkylation of vinylarenes has been demonstrated, proceeding through a radical pathway. A range of valuable (amino)alkyl fragments can be appended onto an alkene substrate to convert abundant feedstocks into valuable linear products. Importantly, electronically diverse vinylarenes are tolerated, representing a breakthrough from the reliance on highly polarized electron-poor alkene substrates for similar reactions. Mechanistic studies have provided valuable insight into the mode of proton transfer in the reaction. Future work will focus on the development of alkene difunctionalization reactions utilizing other radical capture reagents.

## Methods

**Representative procedure for the hydroalkylation of vinylarenes using organotrifluoroborate salts**. In a nitrogen-filled glove box, an oven-dried 4-mL

reaction vial equipped with a stir bar was charged with [Ir(dF(CF$_3$)ppy)$_2$(dtbbpy)]PF$_6$ (2.3 mg, 2 μmol, 1 mol %) and potassium benzyltrifluoroborate (59.4 mg, 0.30 mmol, 1.5 equiv). This was followed by the addition of anhydrous DMF (2000 μL, 0.10 M), 1,1-diphenylethylene (35.3 μL, 0.20 mmol, 1.0 equiv), and guaiacol (67 μL, 0.60 mmol, 3.0 equiv). The vial was sealed with a Teflon-lined cap, removed from the glove box, and irradiated with two 24 W blue LEDs with stirring at 800 r.p.m. at room temperature (r.t.) for 48 h. The crude mixture was then directly adsorbed onto diatomaceous earth (Celite®) and purified by flash column chromatography on silica (2% ethyl acetate in hexanes). Product **1** was obtained as a colorless oil (38.5 mg, 71%).

**Representative procedure for the hydroalkylation of vinylarenes using alkyl carboxylic acids**. In a nitrogen-filled glove box, an oven-dried 4-mL reaction vial equipped with a stir bar was charged with [Ir(dF(CF$_3$)ppy)$_2$(dtbbpy)]PF$_6$ (1.2 mg, 1 μmol, 1 mol %) and K$_3$PO$_4$ (31.8 mg, 0.15 mmol, 1.5 equiv). This was followed by the addition of 1000 μL of a stock solution of cyclohexane carboxylic acid (0.15 mmol, 1.5 equiv) in anhydrous DMF, and 1,1-diphenylethylene (17.6 μL, 0.10 mmol, 1.0 equiv). The vial was sealed with a Teflon-lined cap, removed from the glove box, and irradiated with two 24 W blue LEDs with stirring at 800 r.p.m. at r.t. for 24 h. The reaction mixture was diluted with ethyl acetate (2.5 mL) and then washed with water (3 × 2 mL) to remove most of the DMF. The organic layer was sampled, and the solvent was removed under a vacuum. Product **3** was obtained in 99% yield by flash column chromatography on silica.

**Representative procedure for the hydroaminoalkylation of vinylarenes using α-silyl amine reagents**. In a nitrogen-filled glove box, an oven-dried 4-mL reaction vial equipped with a stir bar was charged with [Ir(ppy)$_2$(dtbbpy)]PF$_6$ (0.9 mg, 0.001 mmol, 0.5 mol %), 1,1-diphenylethylene (35.3 μL, 0.2 mmol, 1.0 equiv), 1-((trimethylsilyl)methyl)piperidine (37.6 mg, 0.22 mmol, 1.1 equiv), and 2.0 mL anhydrous DMF. The vial was sealed with a Teflon-lined cap, removed from the glove box, and irradiated with one 24 W blue LED at r.t. for 3 h. The reaction crude was quenched by the addition of DCM, concentrated in vacuo, and then purified by basic alumina chromatography to afford the desired product **26** in 90% yield.

## Data availability

The data that support the findings of this study are available within the article and its Supplementary information files. Additional data are available from the corresponding author upon request.

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

## Acknowledgements
We thank the NIH (1R35GM125029), NSF CAREER (1555337), the Sloan Research Foundation, Novartis, Eli Lilly, Amgen, and the University of Illinois for their generous support of this work.

## Author contributions
Z.W., S.N.G. and K.L.H. conceived and designed the project. Z.W. and S.N.G. performed the experiments and analyzed the data. Z.W. and S.N.G. wrote the manuscript with contributions from all authors. K.L.H. directed the project.

## Competing interests
The authors declare no competing interests.
