## [Peer Review File · Nature Communications]

REVIEWER COMMENTS

Reviewer #1 (Remarks to the Author):

In this manuscript, Hull and coworkers present a photoredox-based approach to accomplish the anti-Markovnikov hydroalkylation of vinylarenes. This is clearly a valuable reaction—the authors do an excellent job contextualizing their advance relative to prior work. While there are other ways to make these products, this new method is clearly complementary. I particularly appreciate that a range of radical precursors (BF₃K salts, carboxylic acids, alpha-amino C(sp²)-H bonds) each participate readily in this transformation. Additionally, a range of styrene derivatives (EWG, EDG, alpha- and beta-substituted, etc) readily engage in this reaction. However, I do note that many of the examples where the radical component is varied employ 1,1-diphenylethylene as a radical trap. This is clearly a much better vinylarene radical trap than most vinylarenes and, accordingly, they are much easier to functionalize using this new method (for example, compare entry 26 to 41 in both reaction time and yield). If any of the alpha-amino radical precursors were tested with styrene or a simpler vinyl arene but not presented (due to low yield), I'd appreciate seeing these entries included in the SI. Generally, however, the SI was thorough and contained sufficient experimental detail to reproduce these experiments and verify the identity and purity of the compounds produced.

I found the mechanistic experiments reasonably convincing that an electron transfer/proton transfer (ET/PT) mechanism is involved—specifically the d₇-DMF and MeOD experiments. The interpretation of the acid/base additive experiment was not clear to me as the yield change upon addition of AcOH was relatively minor and it was not clear that the carbonate base should be meaningfully deprotonating MeOH under these conditions and it seems that it could interfere with the reaction through a variety of alternative mechanisms. The deuteration experiments more clearly rule out a simple HAT mechanism under these conditions. While the authors show these experiments lead to similar conclusions for the carboxylic acid precursors, I was hoping they could shed some light on the RBF₃K salts given that under those conditions they optimized to include a phenol additive that was rationalized as a HAT catalyst. Can this simply be replaced with a carboxylic acid? Furthermore, the authors note at the outset that reduction of the benzylic radical is thermodynamically demanding. Can the origin of the electron be more clearly identified for the proposed ET/PT mechanism? Finally, I was left curious whether the authors saw any deuteration of the amine fragment in the MeOD experiment (although I do not feel this is a crucial point for the authors to address).

I have one minor concern: the redox auxiliary approach employed to render the hydroaminoalkylation more general was well-described from a practitioner perspective, however, this TMS redox-auxiliary approach was extensively developed by the electrochemistry community and is summarized well in this review: *Chem. Rev.* 2008, 108, 2265–2299. I'd like to see this review, or the seminal papers in this area, cited in addition to the more recent work exploiting this strategy in photoredox catalysis (such as Ref 57).

Overall, I find this manuscript likely to be interesting to the broad readership of *Nature Communications* and recommend this manuscript for publication with minor revisions according to the concerns outlined above.

Reviewer #2 (Remarks to the Author):

The authors describe a photoredox-catalyzed hydro(amino)alkylation of styrene derivatives from either carboxylic acids, BF₃K salts or aminosilanes. The general premise behind this work is that a carbon-based radical is generated from one of the three aforementioned radical precursors using a photoredox catalyst that undergoes a Giese addition in anti-Markovnikov fashion followed by a hydrogen atom transfer to give the final adducts. A range of hydro(amino)alkylated styrenes is synthesized via this reaction and the mechanism is mostly plausible (though H-atom transfer is the most likely final step of the catalytic cycle rather than reduction/protonation).

As for the innovation, there really is none to speak of in this submission. Carboxylic acids, BF₃K salts and aminosilanes have been utilized as radical precursors by multiple groups including MacMillan, Yoon,

Nicewicz, Molander and others in many types of radical reactions including Giese additions to Michael acceptors and in some cases, with absolute stereocontrol. The omission of the large body of this work in the discussion and the references section is both egregious and disingenuous. The only new work here is the use of styrenes, which is viewed as incremental advance at best. The products of this transformation are mildly interesting at best and there are certainly far better disconnections to make these adducts from alternative starting materials. This work is better suited for a more specialized organic chemistry journal such as J. Org. Chem. or Eur. J. Org. Chem.

Reviewer #3 (Remarks to the Author):

Visible light-driven photocatalytic alkene functionalization is an area of intense research in organic chemistry. In this manuscript, Hull and co-workers take advantage of potassium trifluoroborate salts, carboxylic acids and tertiary amines as alkyl radical precursors to accomplish anti-Markovnikov hydro(amino)alkylation reaction through visible light-driven photoredox catalysis. Under the optimized conditions, a wide range of electronically diverse vinylarenes are well tolerated. This protocol could also be extended to alpha-TMS tertiary amines. Successful application of this methodology to several natural product derivatives also highlights its synthetic utility. The corresponding anti-markovnikov hydro(amino)alkylation products are obtained with moderate to good yields. Key to the success of this protocol is the photocatalytic regioselective addition of carbon radical to vinylarenes and identification of efficient HAT reagents. A series of control experiments also support the mechanistic hypothesis. Considering readily available radical precursors, practical conditions, and significance of the products, I think this protocol should be of great interest to the synthetic community.

Moreover, the manuscript is well written, and the supporting information is thorough and provides all the expected data. In conclusion, this is a nice paper and can be accepted in Nature Communication. I do not have any technical issues as the work is scientifically sound.

Minor points:

- 1) The R group of product 5 should be indicted in Table 2.
- 2) Please check the format of references.

Kami L. Hull
Associate Professor of Chemistry
University of Texas at Austin
Department of Chemistry
100 E. 24th Street
Austin, TX 7872
Email: kamihull@austin.utexas.edu
Phone: (734)-417-6009

May 20, 2021

Dear Reviewers,

Thank you for reviewing our manuscript. The following are our responses and efforts to the your comments:

I. Re: referee 1

We appreciate the kind comment of referee 1 on our work: *“This is clearly a valuable reaction—the authors do an excellent job contextualizing their advance relative to prior work. While there are other ways to make these products, this new method is clearly complementary.” and “Overall, I find this manuscript likely to be interesting to the broad readership of Nature Communications and recommend this manuscript for publication with minor revisions according to the concerns outlined above.”*

(a) Original comments: *“However, I do note that many of the examples where the radical component is varied employ 1,1-diphenylethylene as a radical trap. This is clearly a much better vinylarene radical trap than most vinylarenes and, accordingly, they are much easier functionalize using this new method (for example, compare entry 26 to 41 in both reaction time and yield). If any of the alpha-amino radical precursors were tested with styrene or a simpler vinyl arenes but not presented (due to low yield), I’d appreciate to seeing these entries included in the SI.”*

We agree with the reviewer that 1,1-diaryl ethylene is a better carbon-centered radical trapper than simple styrenes. Indeed, as the reviewer mentioned, by comparing the yield and reaction time, the reactivity trend appears to be: 1,1-diaryl ethylene ~ electron-deficient styrene > electron-neutral styrene > electron-rich styrene. This is consistent with the polarity-matched radical addition principle. Moreover, electron-withdrawing groups further increase the reduction potential of the resulting benzylic radical, which facilitates the single electron transfer from the reduced ground state photocatalyst (Ir^{II} in Scheme 3) to form the benzylic anion.

We have also included two unsuccessful substrates in Supplementary Table 19.

Two unsuccessful example:

(b) Original comments: *“The interpretation of the acid/base additive experiment was not clear to me as the yield change upon addition of AcOH was relatively minor and it was not clear that the carbonate base should be meaningfully deprotonating MeOH under these conditions and it seems that it could interfere with the reaction through a variety of alternative mechanisms.”*

We agree with reviewer I on this point. There are a range of mechanisms and effects by which these additives could change reaction performance, and they would be difficult to precisely pinpoint. Additionally, photoredox reactions are more susceptible to perturbations in photon flux, which is a further confounding variable when considering the reaction with Cs_2CO_3 . Given the lack of clarity in these studies, we think this result is best removed from the manuscript. We note that this does not change the overall mechanistic framework of the work.

(c) Original comments: *“While the authors show these experiments lead to similar conclusions for the carboxylic acid precursors, I was hoping they could shed some light on the RBF3K salts given that under those conditions they optimized to include a phenol additive that was rationalized as an HAT catalyst. Can this simply be replaced with a carboxylic acid? ”*

We thank reviewer I for this kind suggestion. Indeed, we did try acetic acid as the H-source, and the desired product 1 was obtained in 15 % yield. A new entry now is added to the Table 1 to provide this insight to the prospective reader.

Based on our mechanistic studies, we believe the ET/PT mechanism is more likely when carboxylic acids and alpha-TMS amines are used with diaryl alkenes or electron deficient monoaryl alkenes. As for RBF₃K salts, as electron rich phenols were the optimal H-source, a HAT mechanism might well be occurring in this case.

(d) **Original comments:** *“Furthermore, the authors note at the outset that reduction of the benzylic radical is thermodynamically demanding. Can the origin of the electron be more clearly identified for the proposed ET/PT mechanism?”*

This is an excellent point, and one we have pondered over for quite some time. Indeed, reviewer I is correct that reduction of the benzylic radical is thermodynamically challenging. To put this in perspective, the reduction potential for the styrene-derived radical ($E_{1/2} = -1.60$ V vs. SCE) is well out of range of the reduced photocatalyst ($E_{1/2} = -1.37$ V vs. SCE). This rules out the sequential ET/PT mechanism for the electron-neutral and -rich monoaryl alkene substrates presented in the manuscript. On the other hand, a direct H-atom abstraction from the MeOH donor is also thermodynamically unfeasible based on bond-dissociation energy considerations. The BDE for a benzylic C–H bond is around 85 kcal/mol and the BDE for MeOH is around 107 kcal/mol. To reconcile this, we favor a PCET mechanism in these cases where the thermodynamics are mismatched. The combination of MeOH ($pK_a = 29$) and the Ir photocatalyst ($E_{1/2} -1.58$ vs. Fc) furnishes a BDFE of 58 kcal/mol, which is much lower than that of the benzylic C–H (85 kcal/mol), thereby rendering the step thermodynamically feasible.

However, this mechanism is not a catch-all for every substrate we have presented. The mechanism for the analogous diarylalkene substrates presented could very well be different. The reduction potentials here are $E \approx -1.34$ V vs. SCE. This is now within the range of the photocatalyst. Therefore, a sequential ET/PT mechanism could be operative for this class of substrates.

To address this reviewer comment, we have added a brief discussion near the end of the mechanism section of the manuscript.

(e) **Original comments:** *“Finally, I was left curious whether the authors saw any deuteration of the amine fragment in the MeOD experiment (although I do not feel this is a crucial point for the authors to address)”*

We thank reviewer I for this question. Indeed, we didn't observe deuterium incorporation into the amine fragment in the MeOD experiment, as indicated by both 1H and 2H NMR of the product. These studies and spectrums were included in the “Mechanistic Studies” section of the SI.

^1H NMR of product

^2H NMR of product

(f) Original comments: *“I have one minor concern: the redox auxiliary approach employed to render the hydroaminoalkylation more general was well-described from a practitioner perspective, however, this TMS redox-auxiliary approach was extensively developed by the electrochemistry community and is summarized well in this review: Chem. Rev. 2008, 108, 2265–2299. I’d like to see this review, or the seminal papers in this area, cited in addition to the more recent work exploiting this strategy in photoredox catalysis (such as Ref 57).”*

We thank reviewer I for this kind suggestion and we completely agree that these early studies should be mentioned in the manuscript. We’ve added following sentence in the TMS-amine section: “This redox auxiliary approach

has been extensively studied in modern electroorganic synthesis.⁶¹ where this Chem Rev paper was cited as ref 61.

II. Re: referee 2

(a) **Original comments:** *“A range of hydro(amino)alkylated styrenes is synthesized via this reaction and the mechanism is mostly plausible (though H-atom transfer is the most likely final step of the catalytic cycle rather than reduction/protonation)”*

For hydroalkylation with carboxylic acid and hydroaminoalkylation with alpha-TMS amines, we believe that the reduction/protonation mechanism is more likely based on the deuterium studies in Scheme 5 and SI (page S36). Moreover, hydrogen atom abstraction of either carboxylic acid or MeOH by the resulting benzylic radical from is thermodynamically unfavored.

As for RBF3K salt, a HAT mechanism might occur in this case as phenols are good hydrogen atom donor.

(b) **Original comments:** *“Carboxylic acids, BF3K salts and aminosilanes have been utilized as radical precursors by multiple groups including MacMillan, Yoon, Nicewicz, Molander and others in many types of radical reactions including Giese additions to Michael acceptors and in some cases, with absolute stereocontrol. The omission of the large body of this work in the discussion and the references section is both egregious and disingenuous. The only new work here is the use of styrenes, which is viewed as incremental advance at best.”*

We don't believe this is a fair comment. We have already stated clearly in the manuscript that **“Anti-Markovnikov-selective hydro(amino)alkylations of electron deficient alkenes initiating with the addition of a carbon-centered radical to the π -system have been well-developed. However, given the nucleophilic character of (amino)alkyl radicals, such reactions have largely been restricted to highly polarized alkenes, such as Michael acceptors. ^{37-41”}**

Seminal work from Giese (ref 39), as well as works from Nishibayshi (ref 40), MacMillan (ref 41), Baran (ref 42), Nicewicz (ref 43) were originally cited. However, to address this reviewer concern, we have added representative work from Yoon (ref 44) and Molander (ref 45) in the revised manuscript. In these pioneering examples, Michael acceptors or polyfluorinated alkenes are used as the trap of carbon-centered radicals. We believe that expanding the scope to more general aryl alkenes will be attractive to the synthetic community.

III. Re: referee 3

We appreciate the kind comment of referee 3 on our work: *“the manuscript is well written, and the supporting information is thorough and provides all the expected data. In conclusion, this is a nice paper and can be accepted in Nature Communication. I do not have any technical issues as the work is scientifically sound.”*

(a) Original comments: *“The R group of product 5 should be indicted in Table 2.”*

Thanks for pointing it out. The corresponding R group information has been added.

(b) Original comments: *“Please check the format of references.”*

All the references have been checked and reformatted as per the Nat. Commun. style.

IV. Re: editor

(a) Original comments: *“In addition to the technical issues raised by Reviewer 1, I believe the work of Knowles et al. (Science 2017, 355, 727–730) should be discussed along with the literature expansions that Reviewers 1 and 2 recommend.”*

We thank the editor for this kind suggestion. Indeed, the aliphatic amine products from hydroaminoalkylation reactions can be made alternatively via hydroamination albeit by a different disconnection. We have modified the manuscript to include the radical hydroamination work pioneered by Knowles (ref 37, 38).

In summary, we feel all the suggestions and concerns from the referees have been carefully addressed or explained with experimental support. Finally, thank you very much again for the time you spend reviewing our manuscript.

Sincerely,

Kami L. Hull

REVIEWERS' COMMENTS

Reviewer #1 (Remarks to the Author):

I am mostly satisfied by these revisions. I agree that there are likely multiple kinetically relevant mechanisms under these conditions and the major contributor likely changes with the substrate. I feel the revised mechanistic discussion is appropriate but could be slightly improved (see below).

I would recommend that a review of Knowles' work be included when suggesting a PCET mechanism may be active (Acc. Chem. Res. 2016, 49, 8, 1546–1556 would be appropriate). That said, I'm not entirely convinced a simpler ET/PT mechanism can be ruled out here. I fully acknowledge I myself questioned the ET/PT mechanism at the outset but, given the mechanistic complexity, I also don't think it can be fully excluded and a slightly more nuanced discussion would be merited.

There are examples of mediated electrolysis undergoing endergonic electron transfer of ~ 590 mV as long as there is a sufficiently rapid way to irreversibly trap the uphill reduced/oxidized intermediate. Protonation of a carbanion certainly fits that bill. I would recommend softening of the language regarding potentials that was added starting at 237 to avoid running afoul of this body of literature.

I also find it a little strange that initially the authors consider a HAT pathway and find phenols to be uniquely effective in their optimization with the organoboron reagents and then perform mechanistic studies on the other system and conclude ETPT in this other context. However, I recommend leaving how this incongruence is addressed (or whether it is) entirely at the author's discretion as I do not think it dramatically impacts the scientific conclusions of the manuscript and manuscripts can always be "more complete" but authors deserve to move on when they are ready.

Kami L. Hull
Associate Professor of Chemistry
University of Texas at Austin
Department of Chemistry
100 E. 24th Street
Austin, TX 7872
Email: kamihull@austin.utexas.edu
Phone: (734)-417-6009

June 30, 2021

Dear Reviewers,

I. Re: referee 1

(a) Original comments: *"I would recommend that a review of Knowles' work be included when suggesting a PCET mechanism may be active (Acc. Chem. Res. 2016, 49, 8, 1546–1556 would be appropriate)."*

We thank the reviewer for this kind suggestion. Knowles' PCET review has been cited in the mechanism discussion as Ref 64.

(b) Original comments: *"That said, I'm not entirely convinced a simpler ET/PT mechanism can be ruled out here. I fully acknowledge I myself questioned the ET/PT mechanism at the outset but, given the mechanistic complexity, I also don't think it can be fully excluded and a slightly more nuanced discussion would be merited."*

We do agree with reviewer that a stepwise ET/PT mechanism cannot be completely ruled out based on our mechanistic studies. We have modified the mechanism discussion to address this concern.

(c) Original comments: *"There are examples of mediated electrolysis undergoing endergonic electron transfer of ~590 mV as long as there is a sufficiently rapid way to irreversibly trap the uphill reduced/oxidized intermediate. Protonation of a carbanion certainly fits that bill. I would recommend softening of the language regarding potentials that was added starting at 237 to avoid running afoul of this body of literature."*

We thank the reviewer for the suggestion. We have modified the mechanism discussion as follows: "Although slightly endergonic, this electron transfer could be kinetically feasible if cage escape and protonation of the subsequently formed carbanion competes with back electron transfer. Alternatively, a concerted ET/PT mechanism could render this step thermodynamically feasible. Although either mechanism cannot be strictly

refuted based on current evidence, it is likely that different mechanisms may be operative for different substrates based on electronic properties.”

(d) Original comments: *“I also find it a little strange that initially the authors consider a HAT pathway and find phenols to be uniquely effective in their optimization with the organoboron reagents and then perform mechanistic studies on the other system and conclude ETPT in this other context. However, I recommend leaving how this incongruence is addressed (or whether it is) entirely at the author’s discretion as I do not think it dramatically impacts the scientific conclusions of the manuscript and manuscripts can always be “more complete” but authors deserve to move on when they are ready.”*

We thank the reviewer for point this out. Indeed, we did start with the organoboron reagents and found out phenols are effective HAT reagents for hydroalkylation of styrenes. Later, we found that broader scope (Fig 3, hydroalkylation section) was obtained when readily available carboxylic acids were used as substrates for the same transformation, without adding any external H-sources. Thus, for hydroalkylation section, we focused more on carboxylic acids for both substrate scope and mechanism. Mechanistic studies with carboxylic acids were carried out as well (See Supplementary Information, P39-41). Similar results were obtained as the hydroaminoalkylation system (Fig. 6).

Finally, we thank the reviewer very much again for the time and kind support.

Sincerely,

Kami L. Hull